# Stem Cells and Acellular Preparations in Bone Regeneration/Fracture Healing: Current Therapies and Future Directions

**DOI:** 10.3390/cells13121045

**Published:** 2024-06-17

**Authors:** Marcel G. Brown, Davis J. Brady, Kelsey M. Healy, Kaitlin A. Henry, Ayobami S. Ogunsola, Xue Ma

**Affiliations:** 1Wake Forest University School of Medicine, Winston-Salem, NC 27157, USA; 2Department of Orthopaedic Surgery and Rehabilitation, Wake Forest University School of Medicine, Winston-Salem, NC 27157, USA

**Keywords:** stem cells, bone regeneration, fracture healing

## Abstract

Bone/fracture healing is a complex process with different steps and four basic tissue layers being affected: cortical bone, periosteum, fascial tissue surrounding the fracture, and bone marrow. Stem cells and their derivatives, including embryonic stem cells, induced pluripotent stem cells, mesenchymal stem cells, hematopoietic stem cells, skeletal stem cells, and multipotent stem cells, can function to artificially introduce highly regenerative cells into decrepit biological tissues and augment the healing process at the tissue level. Stem cells are molecularly and functionally indistinguishable from standard human tissues. The widespread appeal of stem cell therapy lies in its potential benefits as a therapeutic technology that, if harnessed, can be applied in clinical settings. This review aims to establish the molecular pathophysiology of bone healing and the current stem cell interventions that disrupt or augment the bone healing process and, finally, considers the future direction/therapeutic options related to stem cells and bone healing.

## 1. Introduction

The term ‘stem cell’ (*stammzelle*) was first introduced into the scientific community in the late 19th century by zoologists Theodor Boveri and Valentin Häcker, who proposed the existence of a universal precursor cell for both primordial germ and somatic cells [1,2,3,4]. Over the past century, our understanding of stem cells has evolved, culminating in their definition as unspecialized cells with an inherent capability for differentiation and self-renewal [5]. Given their aptitude for regeneration, stem cells have shown tremendous promise in clinical therapy. The first use of stem cells as therapy occurred nearly 70 years ago, when the first bone marrow transplant was administered in 1956 [6]. Since then, stem cells have been extensively applied and studied in numerous medical specialties, including orthopedics, where they have been a major focus of research in the context of bone healing.

It has been suggested that up to 10% of fractures that occur heal incorrectly, leading to an imperfect union [7,8,9]. Extensive research has revealed the critical role of the immune microenvironment in influencing the outcomes of bone healing and regeneration processes [10,11]. Claes et al. performed an in vivo study investigating differences in the cytokine environment between mice with a singular tibial fracture, multiple fractures, or multi-fracture and soft tissue injury. They found that increasing the severity of the injury increased IL-6 levels one day post-injury and halved the load to failure in mechanical tests with reduced callus volume after 28 days of healing [12]. This experiment highlights the multiplying effect of injury and the need for optimal functioning of the tissues surrounding a fracture in order to obtain the most effective healing of fractures. Understanding the physiology and pathology of fracture healing at the molecular level may provide insight into therapeutic measures to minimize fracture healing failure. Stem cells provide a promising solution to this problem; however, creating an environment that enables stem cells to differentiate correctly and promote regeneration remains a challenge.

This manuscript aims to offer a comprehensive overview of stem cell technology in fracture healing and management, including current applications, potential risks, and complications. It categorizes various stem cells and their roles in bone repair and summarizes the current in vitro and in vivo research on bone regeneration. Additionally, it explores future directions and therapeutic applications of stem cells in enhancing fracture healing.

## 2. Clinical Background on Bone Healing

Fracture healing is a well-studied topic in orthopedic research. All mechanisms of bone healing involve four basic tissue layers: cortical bone, periosteum, fascial tissue surrounding the fracture, and bone marrow [13]. Generally, there are two principal mechanisms by which a bone undergoes healing after injury: primary and secondary bone healing. Primary bone healing occurs when a fracture is managed via open reduction and internal fixation, leading to a highly stabilized fracture site [14]. First, the edges of the bone undergo direct appositional bone growth, followed by the small gaps rapidly filling with mature lamellar bone and the large gaps slowly filling with primitive woven bone [15]. Woven bone requires further remodeling and undergoes a process known as contact healing. During this stage, a “basic remodeling unit” forms a cutting cone, which includes peripheral osteoclasts that create a cavity across the fracture, allowing the vasculature and mesenchymal stem cells (MSCs) to enter the tunnel. MSCs eventually differentiate into osteoblasts, which then restore the lamellar bone to the fracture site [15].

In contrast, secondary healing is a more common form of bone healing and is observed when there is an increased level of mobility, leading to micromotion at the fracture site [16]. This type of fracture is often observed in less stable fixation techniques such as splinting, external fixation, and intramedullary nailing. Secondary healing can be divided into five temporal stages. The first stage involves the formation of a hematoma, which is created from the bleeding bone [17]. Inflammation occurs following hematoma formation, which is essential for the recruitment of the necessary molecular components for bone healing. Many important signaling molecules and their functions were summarized by Phillips in 2005, including interleukin 1 (IL-1), interleukin 6 (IL-6), transforming growth factor β (TGF β), insulin-like growth factor (IGF), fibroblast growth factor (FGF), platelet-derived growth factor (PDGF), and bone morphogenetic proteins (BMPs) [18]. Without these molecules, the bone cannot properly heal itself [18]. Inflammation also leads to angiogenesis at the fracture site, which is essential for osteogenesis [16]. Once the site is vascularized, a cartilaginous callus begins to form, and the cartilage eventually calcifies into the bone. Finally, some of the bone is degraded and remodeling occurs via osteoclasts [18]. Sometimes, these five stages are simplified into three phases: inflammation, repair, and remodeling [16].

Bone fractures are among the most frequent traumatic injuries encountered in the emergency department [19,20]. It has been estimated that there may be three million fractures in 2025 due to the increasing number of elderly patients in the United States [21]. The orthopedic field has taken an interest in stem cells as they show potential benefits for bone healing and regeneration.

## 3. Stem Cells

Stem cells are a unique type of cell that have the ability to self-renew and differentiate into a variety of cell lineages [6]. The ability of stem cells to differentiate is deemed potency. Totipotent stem cells are the most versatile type of stem cells and are capable of developing into any cell type in a given organism (both extra-embryonic and embryonic tissue) [6]. Totipotent stem cells are the cells observed at the zygote stage of embryogenesis. Soon after zygote formation, these cells differentiate into an inner cell mass and trophectoderm [22]. The inner cell mass consists of the next most potent stem cell, pluripotent stem cells. While pluripotent stem cells alone are unable to become a complete organism, they are capable of differentiating into any of the three embryonic tissue types (endoderm, mesoderm, and ectoderm) [23]. The final three categories include multipotent, oligopotent, and unipotent stem cells, which can differentiate into specific cell lineages, various cell types, and a single cell type, respectively [5]. Within the human body, certain stem cells are inherent to specific organs or tissues; however, these cells are not considered primitive. These include satellite cells in muscles and intestinal stem cells, which possess tissue-specific differentiation capabilities rather than pluripotency. For therapeutic applications, the focus is often shifted to pluripotent stem cells, which are capable of differentiating into various tissue types. Figure 1 illustrates the predominant stem cells currently utilized in the field. Human stem cell types and applications are summarized in Table 1.

### 3.1. Human Pluripotent Stem Cells (hPSCs)

Human pluripotent stem cells are stem cells derived from human tissue that can produce any type of embryonic tissue. These cells are typically grouped into two categories: naturally derived embryonic stem cells and induced pluripotent stem cells (iPSCs).

### 3.2. Embryonic Stem Cells (ESCs)

Embryonic stem cells (ESCs) are pluripotent stem cells extracted from the inner cell mass of a blastula [26]. Neonatal MSCs, on the other hand, originate from the umbilical cord, placenta, and amnion [27,28]. The fact that ESCs can self-renew at a high capacity, coupled with their ability to differentiate into embryonic tissues, makes them an extremely attractive cell type for potential regenerative therapies. However, major obstacles remain to the successful implementation of ESCs in clinical practice. One challenge with ESCs is being able to differentiate them into the desired cell lines [29]. Another complication of ESCs is the risk of them differentiating into cancerous tissues [30]. Several studies have evaluated ESCs and their risk for teratoma formation in both immunocompromised and immunocompetent mouse models [31,32,33,34,35,36,37,38,39]. An additional challenge of allogeneic and autologous ESCs is that they carry high and moderate risks of host immune rejection, respectively. This is particularly true when the cell reaches a differentiated state, as it is more likely to have a higher number of donor major histocompatibility class I (MHC I) proteins on its surface [40]. To make matters more challenging, immune-mediated cell death of donor ESCs often occurs rapidly, preventing the cells from having time to produce their regenerative effect [29]. Finally, and most importantly, there is an ongoing debate regarding whether the use of embryonic stem cells for research is ethical. Much of this discussion centers on what scientifically constitutes a human life and often involves a vast array of differing social, cultural, and religious ideologies [1,41]. The initial concern in regards to embryonic stem cells centers around the destruction of human embryos in order to extract cells from the blastocyst. By 2001, funding for this research was severely limited with no future funding for embryonic stem cells being extended after 9 August 2001 [42]. Much of the restrictions that were imposed in 2001 were lifted in 2009 under a new administration [43], but the debate surrounding the ethics of embryonic stem cell research remains, with continued policy changes and restrictions on fetal tissue research as recently as 2019. The ability to use previously harvested embryos for research in and of itself is a challenging question and has led to diverging opinions. As an alleviation to much of this debate, the discovery of induced pluripotent stem cells (iPSCS) has provided an escape from much of this discussion as they have proven to be just as capable as embryonic tissue in terms of their pluripotency and potential.

Despite a shift away from the use of ESCs, ESC use is still prevalent, with 50 clinical trials on ESCs having been performed between 2011 and 2022 [44]. However, these clinical trials have largely been limited owing to ethical and experimental concerns [45]. Some of these limitations include the development of chromosomal abnormalities when cultured for long periods, failure of ESCs to become mature and functional cells, the aforementioned potential for ESCs to create teratomas, and continual ethical concerns [46,47,48].

In orthopedics, ESCs have shown the potential to induce chondrogenesis and osteogenesis in vitro and in vivo; however, their clinical applications have been sparse. Bielby and Polak showed that osteogenic differentiation of human ESCs into bone tissue was successfully observed via the identification of osteocalcin and Runt-related transcription factor 2 (Runx2), an osteogenic transcription factor. Furthermore, they implanted this tissue into immunocompromised mice using a poly-D, L-lactide scaffold and found evidence of ESC-derived mineralized tissue after 35 days [49]. In 2021, Petrigliano et al. performed xenotransplantation of human ESCs into porcine knee joints and analyzed chondrogenesis. They found that chondrocyte-differentiated ESCs led to better cartilage repair of the knee articular cartilage six months after transplantation. Additionally, they found no difference in the inflammatory infiltrates between the ESC and control groups, indicating a lack of immunogenicity [50].

### 3.3. Induced Pluripotent Stem Cells (iPSCs)

Induced pluripotent stem cells (iPSCs) are a type of engineered stem cell, similar in function to embryonic stem cells (ESCs). Unlike ESCs, these cells are produced from somatic cells and are biologically induced to become pluripotent. The discovery of iPSCs came about in 2006 when it was elucidated by Takahashi and Yamanaka [51]. They discovered that the alteration of four murine transcription factors through a retrovirus can lead to an induced stem cell with pluripotent capabilities nearly identical to those of ESCs. One year later, Yamanaka et al. successfully performed these experiments using dermal fibroblasts [52]. Since then, the bioengineering of iPSCs has increased exponentially, and new gene cell therapies have been used to induce pluripotency, which no longer requires the use of a retrovirus. The recent discovery of CRISPR/Cas9 gene editing has made the genetic editing of iPSCs even easier and more flexible [53].

The advantages of using iPSCs are numerous, but the two most obvious are the relatively lower level of ethical concern compared to the use of ESCs and the autologous nature of iPSC therapy. Given that iPSCs are harvested from somatic cells rather than embryonic cells, the ethical concerns are minimal compared to ESCs [54]. In addition to ethical advantages, iPSCs also provide improvements from an immune rejection standpoint. As mentioned earlier, therapeutic ESCs are necessarily harvested from a non-self-human, which, like any allogeneic transplant, presents a risk of transplant rejection. iPSCs, on the other hand, can be harvested from the patient, engineered to pluripotency, and then reintroduced into the patient’s body. Unfortunately, some studies have found various instances of immune rejection when using iPSC transplantation [55,56]. Coupled with this disadvantage, iPSCs have often been shown to form teratomas [57,58,59]. The teratogenicity of iPSCs warrants further research to elucidate their characteristics on safety and efficacy in clinical applications. [57]. Finally, the induction of iPSCs into the desired cell lineage can be tremendously challenging. One study reported a success rate of less than 1% for the induction of mouse somatic cells [60].

The use of iPSCs in clinical trials has increased significantly since 2016, presumably owing to their advantages over ESCs. From 2017 to 2022, 53 clinical trials involving iPSCs have been performed [44]. Some of the clinical applications studied include cancer, cardiac, and ocular therapies [44]. In the context of bone healing, several studies have examined the effectiveness of iPSC therapy in vivo. In 2022, Zhou et al. isolated bone marrow mesenchymal stem cells from 16 patients with femoral head osteonecrosis and 12 patients with femoral neck fractures. These mesenchymal stem cells were reprogrammed into iPSCs and then differentiated into iPSC mesenchymal stem cells (iPSC-MSCs). When compared in vivo to bone marrow mesenchymal stem cells (BM-MSCs), iPSC-MSCs were morphologically and immunophenotypically similar. They also showed similar DNA methylation patterns. However, the iPSC-MSCs showed a higher level of proliferation. In a pre-clinical in vivo rat femoral head necrosis model, iPSC-MSCs showed equivalent results in comparison to BM-MSCs in regards to bone maintenance and repair, indicating the feasibility of using iPSCs in clinical bone-healing therapies [61].

A similar study was performed by Jungbluth et al., who examined the effects of iPSC-MSCs on osteogenesis in vivo using a mini-pig model. Following radiologic and histomorphometric analysis, they found that iPSC-MSC transplant with calcium phosphate granules (CPG) showed significantly greater amounts of new bone formation compared to transplant with CPG alone after 6 weeks [62]. iPSC-MSCs showed comparable results to bone marrow MSC transplants and autologous bone marrow concentrate transplants [62]. In 2018, Wu et al. observed calcified matrix growth upon the ectopic implantation of iPSC-MSCs into nude mice. Interestingly, they also found a significant increase in angiogenesis around these sites when iPSC-MSCs were co-transplanted with anti-bone morphogenic protein 2 (BMP2) antibodies. They concluded that these antibodies bind to BMP2 receptors on iPSC-MSCs, inducing angiogenesis and promoting greater osteogenesis [63].

### 3.4. Amniotic Stem Cells (ASCs)

A third, less common form of pluripotent stem cells exists, which is derived from gestational tissues (namely the placenta) and has aptly been termed amniotic stem cells (ASCs) [6]. Within the category of ASCs, there are a number of subtypes of cells, including human amniotic mesenchymal stromal cells (hAMSCs), human umbilical cord mesenchymal stromal cells (hUMSCs), and human amniotic epithelial stem cells (hAESCs) [64]. During the first two weeks of embryonic development, blastocysts are created and implanted into the endometrial stroma. Following this, a primordial amniotic cavity is created. A layer of epiblast cells surrounds this cavity, which eventually differentiates into amnioblasts, from which the amniotic epithelial layer and hAESCs are derived [6,65]. In contrast, other amnioblasts are derived from the hypoblast layer and eventually differentiate into amniotic connective tissues. This tissue is the origin of hAMSCs [6,65].

ASCs have several potential advantages over ESCs, iPSCs, and mesenchymal stem cells. First, these cells are often relatively easy to harvest and are obtained from the placenta, which would otherwise be discarded [64]. Similar to iPSCs, ASCs are derived from non-embryonic gestational tissue and thus have fewer ethical concerns [6]. Additionally, ASCs have been shown to have very low immunogenicity, which provides clear advantages for ASCs in potential clinical therapy. The immuno-privilege that ASCs experience is due to their low expression levels of major histocompatibility class I antigen (HLA-ABC) and lack of expression of major histocompatibility class II antigen (HLA-DR), β2 microglobulin, and HLA-ABC costimulatory molecules CD40, CD80, and CD8635 [6,66]. In 2018, a clinical trial involving hAESC transplantation was performed in infants with bronchopulmonary dysplasia. Of the six babies that received the transplant, none showed acute immune rejection [67].

Arguably, the most notable advantage of ASCs is their suspected lack of tumorgenicity. Numerous studies have analyzed the tumor-forming potential of ASCs in vivo and have shown no tumor formation [68,69,70,71]. In addition, Phermthai et al. used karyotype analysis to show that ASCs display high chromosomal stability [72]. Recently, studies have been performed to determine whether ASCs can be used for cancer treatment. Interestingly, they found that ASCs can exhibit both tumor promotion and tumor suppression effects depending on the cytokines they secrete [73]. Another study by Meng et al. similarly examined ASCs in the context of tumor treatment and found that umbilical cord MSCs (hUCMSCs) showed great anti-tumor potential, whereas amniotic membrane MSCs (hAMSCs) showed both inhibitory and excitatory effects [74]. Although there is some evidence that ASCs promote pre-existing tumors, they are still widely recognized as non-teratoma-forming stem cells and are thus attractive therapeutically [75].

In 2019, Mohammed et al. compared in vivo the healing effects of ASCs to what is by some considered the gold standard for stem cell therapy, bone marrow mesenchymal stem cells (BM-MSCs). They induced a surgical defect in the lumbar spines of rats and subsequently injected ASCs or BM-MSCs at the injury site. They found that the rats that received ASC therapy not only showed osteogenesis but also showed that it occurred at a more advanced stage than that seen in BM-MSC therapy [76]. Basile et al. examined the effects of human-ASC transplantation compared to mouse bone marrow stromal cell (mBMSC) transplantation in mice with calvarial bone defects. ASCs were exposed to an osteogenic stem cell medium plus ascorbic acid for ten days before transplantation and then transplanted using a scaffold. Through GFP staining of host cells, they showed that ASCs promoted the incorporation of host cells into the graft. The transplant also produced bone-like tissue in the defect compared to controls. However, it was concluded that ASCs did not themselves promote osteogenic differentiation; rather, it was their recruitment of host cells that led to bone growth [77].

### 3.5. Multipotent Stem Cells

Multipotent stem cells are cells that have the capability to differentiate into any cell type within a given cell lineage [24]. These cells are naturally found throughout the human body and are often harvested from different cell lineages. These cells can be classified into three primary categories: mesenchymal stem cells (MSCs), hematopoietic stem cells (HSCs), and neuronal stem cells [78]. However, our review focuses on MSCs, HSCs, and a more recently discovered skeletal stem cell (SSC) as these are the primary cells implicated in bone healing [79,80].

### 3.6. Mesenchymal Stem Cells (MSCs)

Mesenchymal stem cells (MSCs) are multipotent stem cells that are found throughout the human body. These regenerative cells are constantly working, creating new tissues to replace old or damaged cells. Adult MSCs are derived from adult tissues, such as bone marrow, adipose tissue, peripheral blood, and dental pulp. In clinical research, it has been suggested that adipose tissue MSCs (AT-MSCs) are the most frequently used MSCs [81]. This is likely due to the relative ease of harvesting AT-MSCs, particularly when compared to other commonly used tissues, such as bone marrow MSCs.

Numerous studies have been performed on MSCs and their capability to effectively produce and heal bone in vitro. Much of the discussion of in vitro MSC study centers on the efficacy of BM-MSCs versus AT-MSCs. One study performed by Mohamed-Ahmed et al. compared human BM-MSCs and AT-MSCs from nine donors. Their results showed that AT-MSCs led to a significantly higher number of adipogenesis-related genes as well as lipid vesicle formation [82]. In contrast, BM-MSCs showed a greater propensity for bone and cartilage regeneration as measured by increased alkaline phosphatase (ALP) activity, greater calcium deposits, and increased expression of osteogenic gene markers [82]. These results were corroborated by another study by Im et al., who considered similar metrics when comparing the two types of MSCs. They also found increased levels of alkaline phosphatase in BM-MSCs and a higher number of mineralization nodules in BM-MSCs than in AT-MSCs when stained with Von Kossa staining [83]. Laio and Chen wrote a review article on this topic in 2014 and determined that all seven of the articles they looked at comparing BM-MSCs to AT-MSCs found BM-MSCs to have either greater or equal osteogenic capabilities in vitro [84].

Although the use of AT-MSCs in bone healing does not appear as promising compared to BM-MSCs, there are other factors that may contribute to better differentiation. One study compared AT-MSCs to different types of fat cells, known as dedifferentiated fat cells (DFAT). They found that both stem cell types had greater difficulty differentiating when hydrogen peroxide was introduced to the system, mimicking oxidative stress. In contrast, catalase introduction led to a 2.5-fold increase in osteogenesis in both cell types [85]. Another study by Wu et al. examined the bone regeneration capacity of BM-MSCs vs. AT-MSCs under different culture conditions. They found that AT-MSCs showed higher seeding efficiency and type I collagen expression, whereas BM-MSCs showed greater osteogenic capacity under all conditions [86]. Moreover, culture conditions also affected osteogenesis. Dynamic culturing, which consisted of stirring and perfusing the culture, enhanced both AT-MSC and BM-MSC osteogenic proliferation compared to static culture. This indicates that the process by which MSCs are cultured affects their regenerative capabilities [86].

Ultimately, there is little consensus on the type of multipotent stem cell that is most effective for bone healing based on in vivo studies. Several studies have tested the ability of BM-MSCs and AT-MSCs to induce bone growth in vivo and have shown success in both cell types [87]. Similarly, one review compared BM-MSCs and AT-MSCs and found that half of the six studies reviewed showed increased osteogenesis in BM-MSCs compared to AT-MSCs, whereas half showed equal levels of bone growth [84]. In 2013, Aykan et al. examined the effect of BM-MSC transplantation at a mandibular osteotomy site in sheep after three and six weeks. Mandibles on each side were cut and treated with BM-MSC mixed with PBS on left side and PBS alone on right side. At both extinction time periods, the left hemimandibles showed greater radiodensity and cortical bone growth than the right hemimandibles [88].

### 3.7. Hematopoietic Stem Cells (HSCs)

Hematopoietic stem cells (HSCs) are a unique self-renewing cell type, from which all human blood cells are derived. These two lineages may differentiate into myeloid and lymphoid lineages. Myeloid cells include platelets, neutrophils, macrophages, and osteoclasts, as well as others. Lymphoid cells include T and B lymphocytes, as well as natural killer cells [89]. They can be found throughout the human body, including in the peripheral blood, bone marrow, and even the umbilical cord [90]. Given the high degree of turnover observed in blood cells, nearly one trillion new blood cells are created daily [90].

Therapeutically, HSCs have technically been used since the 1950s, when the first bone marrow transplant was performed by E. Donnall Thomas [91]. Since then, they have been used extensively and their applications have been growing. HSCs are most commonly used in hematologic and immunological disorders [92,93]. Some of these include chemotherapeutics and treatments for devastating immunodeficiencies, such as severe combined immunodeficiency and chronic granulomatous disease [92]. However, the use of HSCs in osteogenic engineering has not been widely studied. HSCs switch to proliferative and differentiative states upon fracture. These cells differentiate into the cells necessary to begin the bone repair process (i.e., osteoblasts, osteoclasts, and endothelial progenitor cells) [94]. HSCs are often characterized by cell surface protein cluster of differentiation 34 (CD34). Cells of this origin have been shown to exhibit osteogenic and angiogenic effects [94]. In 2022, Oliveira et al. reviewed the current literature on HSCs and bone healing. They provided many valuable findings, but most relevantly, HSCs repeatedly showed increased osteogenesis when transplanted in vivo [94].

### 3.8. Skeletal Stem Cells (SSCs)

Skeletal stem cells (SSCs) are another type of multipotent stem cell that can be found naturally in fetal and adult bones or engineered from iPSCs or BMP2-treated human adipose stroma. These cells can be identified by the presence of Podoplanin (PDPN), cluster of differentiation 73 (CD73), and cluster of differentiation 164 (CD164) surface cell markers, along with the absence of cluster of differentiation 146 (CD146) [80]. Chan et al. performed a study in which they transplanted either injured or uninjured human phalanges into the flank of immunodeficient mice to observe bone growth and measure the proliferation of SSCs. After transplantation, the bones grew along the flank of the mice. Upon removal and digestion of the transplant, they found that phalanges with an introduced injury showed higher levels of human SSCs than uninjured samples, indicating that SSCs proliferate in response to skeletal injury, leading to regeneration [80]. Another study examined SSCs in bone healing in diabetic mice. They introduced a high serum level of tumor necrosis factor α to inhibit Indian hedgehog (Ihh) signaling. In mice with lower expression levels of Ihh, less expansion of SSCs was observed, and consequently, impaired healing occurred. When Ihh was introduced back into the fracture via the use of a slow-release hydrogel, and SSC expansion along with bone healing was restored [95].

**Table 1 cells-13-01045-t001:** Stem cell type and applications.

Stem Cell Type		Application
Human Pluripotent Stem Cell (hPSC)	Embryonic Stem Cell (ESC)	Osteogenic differentiation of human ESCs into bone tissue was successfully observed and evidence of ESC-derived mineralized tissue was found after 35 days [49].
		Chondrocyte-differentiated ESCs led to better repair of knee articular cartilage 6 months after transplantation following xenotransplantation of human ESCs into porcine knee joints [50].
	Induced Pluripotent Stem Cell (iPSC)	MSCs were reprogrammed into iPSC-MSCs, which showed higher proliferation, as well as morphological and immunophenotypically similarities to BM-MSCs *. iPSC-MSCs showed equivalent results in bone maintenance and repair in the in vivo rat femoral head necrosis model [61].
		iPSC-MSCs with CPG * significantly increased bone formation in a mini-pig model at 6 weeks, matching BM-MSC and autologous bone concentrate transplant results [62].
		Co-transplanting iPSC-MSCs with anti-BMP2 * antibodies in nude mice increased calcification and angiogenesis, possibly indicating enhanced osteogenesis through BMP2 receptor interactions [63].
	Amniotic Stem Cell (ASC)	ASC therapy led to more advanced osteogenesis compared to BM-MSC therapy in rat lumbar spine injuries [76].
		Human-ASC transplantation compared to mBMSCs * transplantation in mice with calvarial defects promoted host cell incorporation and bone-like tissue formation via scaffold transplant. Bone growth was possibly due to host cell recruitment rather than direct ASC-induced osteogenesis [77].
Multipotent Stem Cell (MSC)	Mesenchymal Stem Cell (MSC)	BM-MSCs with PBS * enhanced radiodensity and cortical bone growth at mandibular osteotomy sites in sheep compared to PBS alone at 3 and 6 weeks [88].
	Hematopoietic Stem Cell (HSC)	HSCs in bone healing increased osteogenesis with HSC transplantation in vivo [94].
	Skeletal Stem Cell (SSC)	Injured and uninjured human phalanges were transplanted into immunodeficient mice, observing bone growth. Injured samples showed higher human SSC levels, indicating SSC proliferation in response to skeletal injury [80].
		Tumor necrosis factor α reduced Ihh * expression in diabetic mice, impairing SSC expansion and bone healing. Introducing Ihh via hydrogel restored SSC expansion and bone healing [95].

* BM-MSCs (bone marrow MSCs), CPG (calcium phosphate granules), anti-BMP2 (anti-bone morphogenic protein 2), mBMSCs (mouse bone marrow stromal cells), PBS (phosphate-buffered saline), Ihh (Indian hedgehog).

## 4. Potential Mechanisms of Stem Cells in Fracture Healing

There are two primary mechanisms by which stem cells potentially heal bones after fractures. The original dogma was that the stem cells were incorporated into the tissue of implantation; similar to other forms of cell differentiation, stem cells receive tissue-specific cues and become bony and regenerate. Recently, more research suggests another possible mechanism, that stem cells themselves are not incorporated into the site of injury; instead, they utilize paracrine signaling to attract host cells by providing the necessary regenerative components, leading to bone growth without having to become engrafted into the new bone.

### 4.1. Direct Osteogenic Formation

For many years, the prevailing thought has been that stem cells differentiate and incorporate directly into human bone during regeneration. This remains a possible mechanism and has been shown to be true in various studies. Geuze et al. studied the incorporation of MSCs into rat and mice spines, analyzing stem cell incorporation into bone via bioluminescence imaging (BLI). They first isolated and cultured MSCs that they harvested from the iliac wings of a goat and then seeded them into a BCP scaffold. Then, they implanted these MSCs into mice subcutaneously and observed successful osteogenesis during BLI analysis after 6 weeks. This suggested that MSCs seeded into a scaffold differentiate and are then incorporated directly into bone growth in mice. After proving the osteogenic capability of the MSC cell line and observing bone growth in mice, they wanted to see the viability of MSC transplantation in rats. They implanted the same MSC lineage into rats ectopically (subcutaneously) and orthotopically (paraspinally) [96]. Interestingly, the same effect was not observed in the rat cohorts. Many of the rats studied showed the presence of MSCs at the two-week mark, but only one showed successful incorporation of MSCs into spinal bone formation by the end of the implantation period [96]. Another study by Bhumiratana et al. aimed to engineer the ramus-condyle unit (RCU) using autologous stem cells in Yucatán mini-pigs. They infused AT-MSCs into a decellularized bone scaffold and cultured it for three weeks. After implantation, successful regrowth of the RCU was observed with the incorporation of MSCs analyzed via µCT scanning and Movat’s pentachrome staining of bone and soft tissues [97].

The successful incorporation of MSCs into bone or a similar substance may depend on the scaffold used for the graft [98]. Generally, bone defects that are 50 mm or smaller can be repaired using autologous bone grafting procedures. However, larger lesions are more difficult to manage [99]. The use of scaffolds in combination with stem cell engineering is a promising alternative to bone induction [100]. Meinel et al. examined the use of BM-MSCs for healing a 4 mm bone defect in the cranium of mice using a silk fibroin-based scaffold. This scaffold is advantageous because its makeup is largely collagen, and 90% of bone consists of type I collagen. In this way, tissue-engineered and human MSC-seeded scaffolds showed new bone growth at implant–host interfaces, as well as extracellular matrix proteins sialoprotein and osteopontin. However, osteocalcin was not detected in this study, indicating that the MSCs did not achieve full osteoblastic maturity [101]. Collins et al. performed a comprehensive review of scaffolds used for bone healing and suggested that successful bone grafts will likely use a variety of different materials in concordance with the mixed composition of natural bone [98]. Ultimately, these studies support the concept that cell integration depends on tissue scaffolds that mimic the desired tissues for stem cell integration. This highlights the challenges of cell integration in that stem cells alone are often insufficient for tissue regeneration. Instead, a medium that represents defective tissue is required for successful incorporation of stem cells.

### 4.2. Paracrine Signaling in Bone Regeneration

MSCs are responsible for formation of mesenchymal lineage cells, including osteoblasts, chondroblasts, adipocytes, and other types of stromal cells. Numerous cytokines have been shown to play an important role in bone growth and regeneration. Some of the key growth factors include Transforming Growth Factor (TGF)-β, Hypoxia-Inducible Factor (HIF) 1-α, Chemokine Receptor Type 4 (CXCR-4), Stromal Cell-Drive Factor (SDF)-1, Runx2, osterix (Osx), osteocalcin (OC), platelet-derived growth factor (PDGF), bone morphogenic protein (BMP), and vascular endothelial growth factor (VEGF) [102,103]. The role of these growth factors in the context of bone regeneration was characterized in depth by Setiawan et al. [16]. These growth factors are stored and released from either the stem cell secretome or extracellular vesicles (EVs) during the inflammatory phase of the fracture healing process. Following fracture, released PDGF promotes proliferation of MSCs and facilitates MSC osteogenic differentiation and migration to the injury site by downregulating PDGFα expression and depressing BMP–Smad1/5/8–Runx2/Osx axis signaling [104]. TGF-β1 also helps to recruit MSCs to migrate through Smad signaling in both the bone formation and remodeling phases for restoring bone homeostasis [105].

It is well established that TGF-β, SDF-1, and BMPs are crucial in recruiting MSCs at the beginning of bone repair process and inducing osteogenic differentiation. FGF is also involved in promoting growth of periosteal cells and bone progenitor cells for bone formation [106]. Recently, in the context of bone regeneration, the concept of secretomes has drawn attention for their paracrine effects in preserving the stem cell niche. MSC secretomes are cell-free particles that provide important cytokines, growth factors, and transcription factors to the injury site to promote bone healing. For example, a secretome may include angiogenic factors such as VEGF and PDGF, which recruit novel blood vessel formation at the site; osteogenic factors such as osteopontin, osterix, osteocalcin, Runx2, and TGF-β, which are involved in bone formation at the site, and a myriad of interleukins that act as immunosuppressors or immunomodulators, leading to or repressing inflammation. The combination of these molecules leads to increased levels of native osteoblast differentiation, and thus, bone formation [16]. Figure 2 provides a brief overview of the paracrine signaling involved in bone regeneration.

Additionally, the function of these immune cytokines and other immune cells in bone healing was explained by Yang and Liu [10]. Many studies have suggested that the dominant role of stem cells is the induction of these cytokines via paracrine signaling during the angiogenesis stage of the inflammatory response to fracture healing. More specifically, it is believed that paracrine signaling is the act of cell-to-cell communication where a cell produces a signal that induces changes in a nearby cell, thus altering its behavior or function. This is a local method of signaling and not one that relays messages at great distances or hematogenously, as in endocrine signaling. Rather, paracrine signaling acts locally and is broken down rapidly such that its effect is localized.

Linero and Chaparro performed an experiment in which they attempted to induce osteogenesis in rabbit mandibles using either AT-MSCs or merely conditioned media, without stem cells, that was obtained from AT-MSCs. Both the conditioned media and AT-MSCs were administered via a Human Blood Plasma Hydrogel. They found that AT-MSCs were detectable at the injection site after three days but could no longer be seen after twelve days. Additionally, their analysis showed that conditioned media showed increased levels of bone regeneration [107]. Both the elimination of AT-MSCs and improved osteogenesis with conditioned media suggest that a paracrine signaling mechanism of bone regeneration is not only sufficient for tissue regeneration but is also superior to direct cell integration [107]. Another study by Osugi et al. similarly examined the effect of conditioned stem cell media on calvarial bone regeneration. Using computed tomography imaging, they found that the conditioned media cohort showed a greater area of bone growth than the stem cell-treated cohort, although this cohort still showed bone growth. This suggests that the paracrine environment produced by stem cells may be more essential than the cells themselves [108]. Similarly, a study performed by Reichert et al. explored bone regeneration capacities of BM-MSCs vs. recombinant human bone morphogenic protein 7 (rhBMP-7) for a 3 cm long resection of sheep tibias. They compared this to the gold standard surgical treatment of autologous bone grafting, which has a myriad of additional surgical complications. Their findings revealed that scaffolds cultured with both MSCs and rhBMP-7 promoted bone regeneration at the resection site, with rhBMP-7 achieving superior bone growth. This was confirmed through X-ray, CT, micro-CT scans, biomechanical tests, and histological analysis [109].

Besides MSCs, skeletal stem cells (SSCs) have also been reported to promote fracture healing in the bone repair process. Salhotra et al. thoroughly reviewed the application of SSCs in vitro and in the animal model for bone regeneration. In a mouse fracture model, SSCs that express CD49f contribute to bone repair, but these cells are absent in the uninjured bone. It was concluded that SSC-driven osteogenesis in mice was mediated by focal adhesion kinase (FAK) to enable the SSCs’ embryonic neural crest identity. In a novel human xenograft model, a unicortical injury on human fetal phalanges displayed an elevated frequency of human SSCs and exhibited increased osteogenic potential in vitro [110].

## 5. Stem Cell Therapy: Current State and Applications

The goal of stem cell therapy is to artificially introduce highly regenerative cells into decrepit biological tissues. The benefit of these cells is that in the right conditions they are molecularly and functionally indistinguishable from standard human tissue. The use of stem cells as a therapeutic to repair damage and regenerate natural tissue has been the focus of many researchers over the past few decades and continues to be a heavily funded endeavor, with the stem cell market reaching nearly 15 billionUSD in 2023 and expected to increase by 11.43% annually [111]. This increase in the stem cell market largely stems from the widespread belief that stem cells are not just functional in theory but also have a role in clinical applications.

The current gold standard for replacing lost bone is an autogenous bone graft. Nearly 500,000 graft surgeries are performed annually. While there are certainly benefits to this therapy, it is also highly limited, given the challenges that come with harvesting the autograft [112]. Over the past few decades, stem cell engineering advancements have allowed for greater clinical applicability and, thus, an increasing number of clinical trials. To date, most clinical trials focused on bone healing have used the BM-MSC lineage, which indicates that they are the current gold standard for therapy [113]. Only one clinical trial has looked at umbilical cord-derived MSCs, and another has analyzed AT-MSCs.

### 5.1. BM-MSC Clinical Trials

Much work has been performed globally assessing the effectiveness of implementing BM-MSCs into bone healing therapies. Theodosaki et al. recently published a review of the bone healing capacity of MSCs in scaffolds and found that each of the fourteen studies they analyzed exemplified superior bone healing capabilities compared to the standard of care when using MSCs [114]. In another recent review by Venkataiah et al., all nine completed clinical trials they reviewed demonstrated adequate or even superior bone healing using stem cell transplants [113].

#### 5.1.1. Spine

Another study by Gan et al. looked at the use of porous beta-tricalcium phosphate-enriched (β-TCP) BM-MSC therapy in the context of posterior spinal fusion. They enrolled 41 patients with either thoracolumbar fractures (TLF) or degenerative disc disease (DDD) and, similarly, began the experiment by harvesting bone marrow from the iliac crest. Enriched MSCs were produced by a cell separator peri-operatively and combined with porous β-TCP and incubated. After the incubation, they were implanted back into the patients. The enrichment technique led to a 4.3-fold increase in the number of alkaline phosphatase-expressing colony-forming units, a good predictor of MSC prevalence. MSC prevalence was observed to be higher in younger patients compared to older patients. Similarly, those with thoracolumbar fractures showed higher levels of MSCs than those with DDD. Ultimately, 34.5 months post-operatively, radiographs showed that 95.1 percent of cases demonstrated good spinal fusion as assessed by two independent orthopedic surgeons. This study indicated that cytotherapy was as effective as the gold standard of autologous bone graft but was simpler, quicker, and carried fewer complications [115].

#### 5.1.2. Face

Redondo et al. performed a pilot clinical trial on nine patients with maxillary cysts. They used BM-MSCs in combination with a BioMax scaffold to try seed bone growth into the cysts. Following a pre-op meeting to ensure compliance to set study criteria, the first surgery was performed on patients, harvesting a spongy maxillary bone sample for MSC and blood samples for the serum scaffold. The BM-MSCs obtained were incorporated into the autologously created BioMax scaffold, expanded, and allowed to differentiate for 3–4 weeks. The contralateral side of the spongy bone area served as a control for individual patients. Four weeks later, the BM-MSC scaffold was surgically introduced to the maxillary cyst. CT scans were used to monitor bone formation in the cyst. Compared to pre-treatment, there was a significant increase in the CT density of the BM-MSC-treated cyst interior. Additionally, no inflammation or adverse effects were observed. The ratio of the CT values after/before treatment was 2.52 ± 0.45. Conversely, the density of the contralateral control area of spongy alveolar bone without treatment exemplified no change with an after/before ratio of 0.99 ± 0.14 [116].

Similarly, in 2018, Gjerde et al. published a clinical trial that analyzed the utility of BM-MSC therapy in maxillofacial bone defect healing. Eleven participants with severe mandibular ridge resorption were selected and given a treatment of BM-MSC and a biphasic calcium phosphate granule scaffold. All 11 patients healed without complications or infections. After 4–6 months, clinical and radiographic analysis of bone healing was performed and revealed an average bone volume increase (n = 14) of 887.23 ± 365.01 mm^3^ (*p* < 0.001). Additionally, the measured bone width mean increase (n = 14) was 4.05 mm (*p* < 0.001). Micro-CT and histology show the successful formation of biphasic calcium phosphate (BCP) granules, indicating successful integration of the BM-MSC scaffold into the bone tissue. Lastly, patient-reported functional outcomes and patient satisfaction were recorded 12 months post-op. All 11 patients were satisfied and experienced no bothersome functional outcome loss [117].

#### 5.1.3. Hip (Total/Revision Hip Arthroplasty)

In 2016, Šponer et al. performed a phase IIa clinical trial in which they treated half of the patients with β-TCP in combination with BM-MSCs and the other half with β-TCP alone. Nine patients in each group were undergoing revision total hip arthroplasty due to bone healing defects. Patients were followed and assessed at 6 weeks and 3, 6, and 12 months using the Harris Hip Score, radiography, and DEXA scans. Radiographs showed successful trabecular remodeling for all nine of the patients treated with BM-MSC therapy compared to only one of the patients treated with β-TCP alone [118]. Two years later, in 2018, Šponer et al. looked at 37 patients with osseous defects undergoing revision hip arthroplasty. Different from their previous study, they injected 19 patients with β-TCP plus BM-MSCs (group A), 9 patients with β-TCP graft material alone (group B), and 9 more patients with only cancellous autografts (group C). Upon following the patients 6 weeks and 3, 6, and 12 months post-operatively, they found no significant difference in bone healing between groups A and C but did find a significant difference between groups B and C [119].

#### 5.1.4. Long Bones

In a phase II clinical trial, Granchi et al. enrolled 39 patients in BM-MSC therapy in order to evaluate whether bone turnover markers were adequate predictors of the regenerative ability of bone marrow-derived MSCs. The patients had either experienced nonunion fractures of long bones or osteonecrosis of the formal head (ONFH) (nonunion n = 26; ONFH n = 13). BM-MSCs were harvested from pelvic bones and sent off for culture. Following this, the BM-MSCs were mixed with biphasic calcium phosphate and implanted into the fracture or necrotic site. Bone formation markers, control radiographs, and clinical controls were each measured before intervention, 12 weeks post-surgery, and 24 weeks post-surgery. The controls served to depict the bone healing progression at each time point. Ultimately, of the 39 patients who participated in the study, 37 of them completed the follow-up, and 33 of the 37 achieved good clinical outcomes following treatment. Additionally, no instances of adverse effects were noticed related to the stem cell treatment. For patients who experienced good outcomes, a collagen biomarker called C-Propeptide of Type I Procollagen (CICP) was notably increased, and a collagen biomarker called C-terminal telopeptide of type I collagen (CTX) was notably decreased in both the malunion and ONFH groups. These findings indicate that these biomarkers may be a useful tool for determining the status of a patient’s healing [120].

In further support of the idea that BM-MSC therapy may have an effect on long bone delayed unions and nonunions is a 2020 study published by Gomez-Barrena et al. They combined BM-MSCs with MBCP+^TM^ (a synthetic bone substitute made of 20% hydroxyapatite (HA) and 80% β-TCP) and injected that into 28 patients with long bone delayed union or nonunion fractures. Similar to many other studies, they assessed bone healing using radiographic analysis. They found that 25 patients who finished a one-year follow-up showed bone consolidation. One patient died 6 months into the study due to unrelated causes, and two patients were deemed failures (one voluntarily dropped out of the study at 3 months, and the other showed nonunion 6 months into the study, requiring further surgical intervention). Further quantification of bone growth using the REBORNE bone healing scale showed increasing means of 0.62 ± 0.08 to 0.78 ± 0.09 to 0.89 ± 0.09 at 3, 6, and 12 months, respectively. In addition to radiographs, clinical assessments were performed and found that after 12 months, all 25 patients indicated a Visual Analogue Scale (VAS) pain score lower than 30/100. Bone biopsies were also taken of the fracture sites 8 months post-op, and bioceramic granules were observed surrounded by multinucleated giant cells labeled with CD86 and TRAP stains [121].

Similarly, Jayankura et al. published a phase I/IIA open-label pilot trial where they assessed the effectiveness of BM-MSC therapy on long bone growth analyzed using the Tomographic Union Score (TUS) and a modified Radiographic Union Score. They also monitored the patients’ overall health status by using the Global Disease Evaluation (GDE), as well as by assessing pain to palpation using VAS. Twenty-two patients with 3–7 month delayed unions of long bones were treated with allogenic BM-MSCs harvested from their own iliac crests. One patient was excluded from analysis for failure to follow protocol. Results showed a GDE score improvement of ≥25% and a TUS score increase of ≥2 was seen in 16 out of 21 patients. The mean TUS increase at 6 months post-op was 3.8 points. Additionally, none of the 21 patients analyzed required revision/rescue surgery by the 6-month post-op timepoint. However, by the 18-month timepoint, 2 out of21 patients required revision surgery [122].

In 2016, Seebach et al. performed a very similar phase I clinical trial, looking at the bone-healing effect of BM-MSCs seeded into β-TCP on proximal humeral fractures. Ten patients underwent BM-MSC aspiration one day preoperatively. During their humeral fixation, they received 12 mL of the BM-MSC solution loaded in situ onto the β-TCP. Results showed that patients all tolerated the BM-MSC aspiration well, with no infection occurring. Additionally, radiological evaluations showed no incidences of secondary dislocations or screw perforations were observed. Clinical outcome evaluation showed an average disabilities of the arm, shoulder, and hand (DASH) score of 52 ± 7.9. For reference, a score of 0 means that there is optimal functioning without limitation, and a score of 100 indicates maximal limitation. However, DASH scores are not perfect predictors of bone healing as they can also be influenced by other anatomical dysfunctions in the region of the fracture. Nonetheless, they showed the ease and feasibility of using BM-MSCs for humeral bone healing [123].

Uniquely, a prospective phase IIa clinical trial published in 2024 by Seebach et al. showed results that contradicted the effectiveness of BM-MSCs for bone healing. They looked at the combination of BM-MSCs with β-TCP once again in the context of proximal humerus fractures. They supplemented open reduction internal fixation (ORIF) surgeries with this autologous stem cell therapy to assess whether there would be a decreased incidence of secondary dislocation. They enrolled a cohort of 94 patients, half of whom received BM-MSCs plus β-TCP, and the other half received only β-TCP. However, after running a statistical analysis on 56 patients, the study was terminated because no statistical differences were seen in secondary dislocations or complications. From this, the authors concluded that BM-MSC transplantation is relatively safe and well tolerated, and thus, it may be useful to continue analyzing BM-MSC therapies in clinical trials. They also suggest that the effects of BM-MSC therapy may be greater on diaphyseal bone defects, but it does not seem to have any statistically significant effect on metaphyseal defects [124]. A summary of BM-MSC clinical trials can be found in Table 2.

**Table 2 cells-13-01045-t002:** BM-MSC clinical trials.

Author	Defect	Transplant	Follow Up	Outcome
Granchi et al. (2019) [120]	Nonunion of long bones (n = 26) and osteonecrosis of femoral head (n = 13)	BM-MSCs + biphasic calcium phosphate	12 and 24 weeks	33/37 achieved good clinical outcomes. No adverse effects were observed. In good outcome patients increased CICP and decreased CTX collagen biomarkers were observed.
Seebach et al. (2016) [123]	Proximal humerus fractures (n = 10)	BM-MSCs + β-TCP	5 visits over 12 weeks	Radiological evaluation showed no secondary dislocations or screw perforations
Gomez-Barrena et al. (2020) [121]	Long bones with delayed union or nonunion. Femur (n = 11), Humerus (n = 4), Tibia (n = 13).	BM-MSCs + MBCP+^TM^ (20% HA and 80% β-TCP)	3, 6 and 12 months	25/28 patients showed consolidation 12 months post-op on radiograph. One died at 6 months, one voluntarily dropped out, one experienced nonunion.
Redondo et al. (2018) [116]	Maxillary cysts (n = 9)	BM-MSCs + BioMax scaffold	2 weeks, 3–4 months, and 6–8 months from the 2nd surgery	The ratio of the CT bone density values after/before treatment was 2.52 ± 0.45 in the experimental group and 0.99 ± 0.14 in the control group.
Gan et al. (2008) [115]	Thoracolumbar fracture (n = 19) and degenerative disc disease (n = 22)	BM-MSC + β-TCP	2 weeks and 1, 3, 6, 12, and 24 months	Radiographs showed 95.1% of patients demonstrated good spinal fusion after 34.5 months
Gjerde et al. (2018) [117]	Maxillofacial bone defect (n = 11)	BM-MSC + biphasic calcium phosphate granule scaffold	12–14 days and 1, 6, 9, and 18 months	Average bone volume increase was 887.23 ± 365.01 mm^3^ (*p* < 0.001) and measured bone width mean increase was 4.05 mm (*p* < 0.001).
Seebach et al. (2024) [124]	Proximal humerus fracture (n = 56)	BM-MSC + β-TCP (n = 28); Only β-TCP (n = 28)	1, 6, and 12 weeks post-op	Study was terminated because no statistical differences were seen in secondary dislocations or complications between BM-MSC group and control group.
Jayankura et al. (2021) [122]	Long bone delayed union fractures: Tibia (n = 8) Humerus (n = 5) Femur (n = 3) Ulna (n = 3) Fibula (n = 2) Radius (n = 1)	BM-MSC	2 weeks and 1, 3, 6, 12, and 24 months	GDE score improvement of ≥25% and a TUS score increase of ≥2 was seen in 16/21 of the patients. The mean TUS increase at 6 months post-op was 3.8 points. Two patients needed revision surgery at 18 months post-op.
Šponer et al. (2016) [118]	Total hip arthroplasty (n = 18)	BM-MSC + β-TCP (n = 9); only β-TCP (n = 9)	6 weeks and 3, 6, and 12 months post-op	Trabecular remodeling was found in all nine BM-MSC-treated patients and only in one control patients.
Šponer et al. (2018) [119]	Revision hip arthroplasty (n = 37)	BM-MSC + β-TCP (n = 19); only β-TCP graft material (n = 9); only cancellous autografts (n = 9)	6 weeks and 3, 6, and 12 months post-op	No significant difference between BM-MSC and cancellous autograft groups but significant difference seen in only β-TCP graft compared to cancellous autografts.

### 5.2. Umbilical Cord MSC (UC-MSC) Clinical Trials

UC-MSCs have not traditionally been widely used as therapies for bone healing. However, Shim et al. performed a unique phase I/IIa clinical trial looking at the safety and efficacy of Wharton’s Jelly UC-MSCs in osteoporotic vertebral compression fracture (OVCF) healing. A total of 14 patients completed the study, 7 being assigned to receive UC-MSCs plus teriparatide (parathyroid hormone 1–34), deemed the combined treatment, and the other 7 receiving teriparatide alone. They analyzed various metrics to assess fracture healing, including the VAS, the Oswestry Disability Index [ODI], the Short Form-36 [SF-36], bone mineral density [BMD], bone turnover (measured by osteocalcin and CTX), dual-energy X-ray absorptiometry [DEXA], and CT scans. Of these metrics, the combined treatment group showed a significant increase in the VAS, ODI, and SF-36 scores compared to the control group. However, bone turnover markers were not significantly different. BMD T-scores of the spine and hip analyzed by DEXA showed an increase in both UC-MSCs and control groups with no statistically significant difference between the two groups. Finally, CT scans of the spines of the UC-MSCs treatment group showed better microarchitecture at baseline, 6 months, and 12 months. Shim et al. concluded that UC-MSC therapy is feasible, tolerable, and may lead to better bone healing [125]. A summary of this UC-MSC clinical trial can be found in Table 3.

**Table 3 cells-13-01045-t003:** UC-MSC Clinical Trials.

Author	Defect	Transplant	Follow Up	Outcome
Shim et al. (2021) [125]	Osteoporotic Vertebral Compression Fracture (n = 14)	UC-MSCs + teriparatide (n = 7); only teriparatide (n = 7)	6 and 12 months	UC-MSC treatment group showed a significant increase in the VAS, ODI, and SF-36 scores compared to the control group. Bone turnover markers were not significantly different between the two groups. BMD T-scores of the spine and hip analyzed by DEXA showed an increase in both UC-MSC and control groups with no statistically significant difference between the two groups. CT scans of the spines of the UC-MSC treatment group showed better microarchitecture at baseline, 6 months, and 12 months.

### 5.3. Adipose Tissue MSC (AT-MSC) Clinical Trials

Another less frequently used clinical therapy for bone healing is AT-MSCs. This is an interesting paradigm given the breadth of in vivo and in vitro research using AT-MSCs for bone healing. Nonetheless, Castillo-Cardiel et al. performed a study in 2017 looking at how AT-MSC therapy works for healing mandibular fractures. They enrolled 20 male patients with mandibular fractures and assigned them to a control group and an experimental group unknowingly. Ten received an intermaxillary fixation surgery alone, and the other ten received fixation alongside AT-MSC therapy. The adipose tissue was harvested from patients 24 h before surgery, and AT-MSCs were isolated and cultured. During the fixation, the control group received an injection of AT-MSCs at the fracture site. Panoramic radiography and CT scans were obtained of normal bone and fractured bone 4 weeks and 12 weeks post-op to assess bone intensity and density. They found that bone quality at 4 weeks was 108.82 ± 3.4 for the AT-MSC group compared to 93.92 ± 2.6 for the control group (*p* = 0.000) using panoramic radiography. Similarly, scores of 123 ± 4.53 vs. 99.72 ± 5.72 (*p* = 0.000) were seen using computed tomography. The increase in bone quality of the AT-MSC group increased when analyzed at week 12. Panoramic radiography showed measurements of 153.53 ± 1.83 vs. 101.81 ± 4.83 (*p* = 0.000), and CT showed 165.4 ± 4.2 vs. 112.9 ± 2.0 (*p* = 0.000) for the AT-MSC vs. control groups, respectively. While the level of bone ossification was similar at 4 weeks, there was a statistically significant 36.48% increase in ossification for the AT-MSC-treated group at 12 weeks post-op. They concluded that AT-MSC therapy induced an ossification rate 2.4 times higher than seen in traditional mandibular fracture treatment [126]. Given that AT-MSC harvesting is less invasive than BM-MSC and seemingly efficacious for bone healing, AT-MSC therapy is an attractive alternative to the gold standard of BM-MSC therapy in the clinical setting. A summary of this AT-MSC clinical trial can be found in Table 4.

**Table 4 cells-13-01045-t004:** AT-MSC Clinical Trials.

Author	Defect	Transplant	Follow Up	Outcome
Castillo-Cardiel et al. (2017) [126]	Mandibular Fractures (n = 20)	AT-MSCs + Fracture Reduction (n = 10); Only Fracture Reduction (n = 10)	4 and 12 weeks post-op	AT-MSC-treated group showed an increase in bone quality indicated by panoramic radiography measurements of 153.53 ± 1.83 vs. 101.81 ± 4.83 (*p* = 0.000), and CT measurements of 165.4 ± 4.2 vs. 112.9 ± 2.0 (*p* = 0.000) for the AT-MSC vs. control groups, respectively. A statistically significant 36.48% increase in bone ossification was seen in AT-MSC-treated group at 12 weeks post-op.

## 6. Future Directions

Stem cell therapy, in the context of bone healing and regeneration, is an exciting and promising field of research. Our understanding of the cellular and biochemical environment during bone healing has grown tremendously over the past twenty years. However, there are still many unanswered questions before these stem cell technologies can be widely implemented in the clinical setting.

First, the need for characterization and standardization of these treatments are increasingly necessary. Liquid chromatography/mass spectrometry has been used in some instances to characterize acellular stem cell populations, which are notoriously the least defined of all stem cell populations [127,128]. Protein microarrays have also been used to define stem cell populations [129]. Mass spectrometry and cytokine arrays have also been helpful tools to characterize the protein content of many acellular preparations of stem cell secretomes, with a variety of findings and inconsistent concentrations of protein within the various secretomes themselves [129,130,131,132]. In order to reinforce the reliability of various stem cell therapies, characterization of both the acellular and cellular components of stem cells is required, with further translation of these findings in pre-clinical studies.

Secondly, the intricacies of the biochemical mechanisms of stem cell healing should be the focus of future studies. One specific area of study that requires clarification is determining which type of MSC is better for in vivo use. There is no consensus on whether in vivo use of BM-MSCs or AT-MSCs leads to greater osteogenesis [133]. The current debate is whether to use live MSC cultures (e.g., cell transplants, grafts) or acellular preparations derived from these cells (e.g., conditioned media, extracellular vesicles) for regenerative purposes. If paracrine signaling is central to regenerative therapy, the cellular content is assumed to be superfluous. More studies on acellular secretome preparations versus cellular stem cell therapies need to be compared side by side in both in vitro and in vivo studies for further verification of efficacy. Understanding the immunomodulatory effect of stem cells and their acellular counterparts may prove vital to appreciating the differences in these therapies.

In summary, understanding the underlying mechanisms and standardizing the development and techniques for using stem cell preparations is vital for establishing a consistent response to therapy. It is difficult to measure the effectiveness of a treatment if the implementation methods are inconsistent. There is an urgent need to fully standardize or at least agree on common methods of stem cell development/technique to achieve consensus in terms of which preparation or approach is best for augmenting bone healing.

## Figures and Tables

**Figure 1 cells-13-01045-f001:**
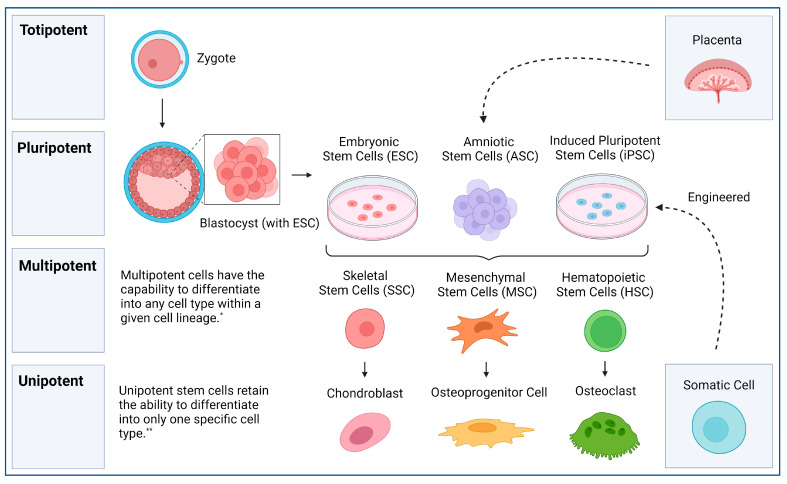
Stem cell origin and lineage. * [24], ** [25] created with BioRender.com.

**Figure 2 cells-13-01045-f002:**
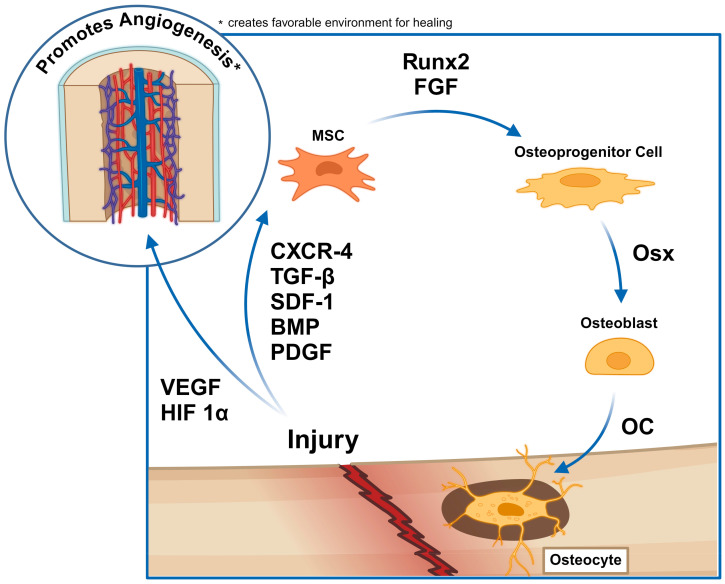
Bone remodeling pathway after fracture. Created with BioRender.com.

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
