# Peer review of "Stem Cells and Acellular Preparations in Bone Regeneration/Fracture Healing: Current Therapies and Future Directions"

_cells, 2024, doi:10.3390/cells13121045_

Round 1
Reviewer 1 Report
Comments and Suggestions for Authors
Dear Marcel G. Brown and co-authors,
Thank you for the very well prepared manuscript entitled "Stem Cells and Acellular Preparations in Bone Regeneration/Fracture Healing: Current Therapies and Future Directions". It was a pleasure to review it. Therefore, I have only minor comments or questions of understanding, which I list below:
1. Lines 363 - 364: For me, the reference to the "Stem Cell Type and Applications" table still comes in the body text of the "Skeletal Stem Cells" chapter. However, as this table deals with different stem cell types, this reference should be separated from the SSCs chapter. Somehow it doesn't fit in this place for me.
2. Lines 382-383: "They implanted MSCs both ectopically (subcutaneously-382 ously) and orthotopically (within the spine), this was only performed in rat cohorts [95]." What was done differently in the mouse experiments than in the rat? In rats it was implanted ectopically and orthotopically and in mice?
3. Line 495-496 "Supplementary Materials: The following supporting information can be downloaded at: 495 www.mdpi.com/xxx/s1, Figure S1: Stem Cell Origin and Lineage." This file could not be found online and was not otherwise available during the review.
Author Response
Thank you for your comments.
- Table 1 is at the end of section 3 Stem cells as a summary for all the stem cells characterized in the text, not only embodied in 3.8 skeletal stem cells (SSCs). We are required to incorporate tables in the text.
- please see the revision line 377-389.
- We have removed the supplemental material as those are figure publication licenses.
Reviewer 2 Report
Comments and Suggestions for Authors
The manuscript delves into the fascinating world of stem cell research, specifically focusing on the role and potential of various types of stem cells in bone healing and regeneration. This critical analysis will follow the provided guidelines, addressing each requested point systematically.
Summary of the Manuscript
The study extensively reviews the advancements in stem cell therapies for bone regeneration, emphasizing the molecular mechanisms, therapeutic applications, and future directions of embryonic stem cells (ESCs), induced pluripotent stem cells (iPSCs), and mesenchymal stem cells (MSCs), among others. It assesses the current state of research, the practical implications of these therapies in clinical settings, and the challenges that need to be addressed to harness the full potential of stem cell treatments for bone healing.
Major Issues
-
Originality and Relevance: The manuscript offers a comprehensive review of stem cell therapies' role in bone healing, a crucial area in regenerative medicine. Its exploration of the mechanistic pathways, clinical trials, and future research directions adds valuable insights to the field . However, while it integrates diverse stem cell types and their applications, the novelty in comparison to existing literature isn't deeply examined, especially regarding the unique contributions of the study to the existing body of knowledge.
-
Methodological Improvements: The document could benefit from a more detailed comparison and meta-analysis of clinical trial outcomes to substantiate its discussions. A systematic approach to evaluating the effectiveness and safety of different stem cell therapies would enhance its methodological rigor .
Minor Issues
-
Tables and Figures: The inclusion of Table 1 summarizing stem cell types and their applications provides a useful at-a-glance overview of the field. However, the manuscript could improve by adding more visual representations, such as flowcharts or diagrams, to elucidate complex mechanisms and the stages of bone healing enhanced by stem cell therapy .
-
Clarity and Explanation: Some sections would benefit from a more detailed explanation for non-experts, such as the differentiation pathways of specific stem cells and their role in bone regeneration. Clarifying these elements could make the paper more accessible to a broader audience .
Consistency with Evidence
The conclusions drawn in the manuscript are generally well-supported by the evidence and studies cited. It effectively links the potential of stem cells with the mechanistic pathways involved in bone regeneration. However, the paper would be strengthened by addressing conflicting results and limitations in the current research landscape more thoroughly, ensuring all conclusions are robustly backed by data .
References
The references are appropriate, covering a wide range of seminal works and recent studies that underpin the manuscript's discussions. However, incorporating more recent studies could further enrich the paper, providing the most up-to-date context for the findings and discussions presented .
Additional Comments
- The quality of data, especially in clinical applications and outcomes of stem cell therapies in bone healing, is crucial. Future iterations could benefit from a deeper dive into the data quality from clinical trials, including patient outcomes, side effects, and long-term follow-up results.
- Ethical considerations, particularly concerning ESCs, are well-noted, but a more nuanced discussion on navigating these ethical waters in future research could add depth to the paper .
In conclusion, the manuscript makes a significant contribution to the field of stem cell research for bone healing, with its extensive review and discussion on the potential of stem cells in regenerative medicine. Addressing the outlined major and minor issues could enhance its impact and utility for both researchers and clinicians in the field.
Author Response
Thank you for your comments.
- Originality and Relevance: We have added more stem cell therapy current state and applications including clinical trails for recent years (section 5) to add to the novelty of the review. Also, more latest findings in regards of potential mechanisms of stem cells have been added to section 4.
- methodological improvements: section 5 clinical trials have been added to improve the rigor.
- tables and figures: figure 2 and graphical abstract have been added to improve visualization
- Clarity and explanation: We have added more details to section 3 and 4 to elucidate the pathways and mechanisms of stem cells in bone fracture healing. However, some of the well established mechanisms were reviewed thoroughly already thus were cited here to avoid redundancy and for the sake of the length of this review.
Reviewer 3 Report
Comments and Suggestions for Authors
In this review, authors describe the therapeutic application of stem cells in the healing and management of fractures and future directions .
The review seems more developed as a review on biological features of stem cells, I think the authors need to add more information about clinical applications.
Authors just make a list without elaborating carefully on how MSCs can improve regeneration.
"Stem Cell Therapy Current State and Applications," sections should be improved, adding some more information, both with in vitro and in vivo studies explaining the mechanisms involved in the regenerative effect of stem cells. In particular some studies highlighting the role of specific molecules in committing AT- MSCs (as for example melatonin and/or vitamin D) toward the osteogenic phenotype in vitro will add more molecular insight to the manuscript.
The use of exosomes derived from MSCs is arising as a novel interesting approach. Authors could add this topic to the future perspectives.
Comments on the Quality of English LanguageMinor editing of English language required
Author Response
Thank you for your comments.
- section 5 stem cell therapy current state and applications has been updated with clinical trials.
- section 4 has been updated with more details of in vitro and in vivo studies for varying stem cells in bone fracture healing.
- exosomes have been elaborated in section 4.2 paracrine signaling in bone regeneration
Round 2
Reviewer 2 Report
Comments and Suggestions for Authors
Following the changes made, the article can now be published in this journal.